# Universal Neural Functionals

**Allan Zhou**
Stanford University
ayz@cs.stanford.edu

**Chelsea Finn**
Stanford University

**James Harrison**
Google DeepMind
jamesharrison@google.com

## Abstract

A challenging problem in many modern machine learning tasks is to process *weight-space features*, i.e., to transform or extract information from the weights and gradients of a neural network. Recent works have developed promising weight-space models that are equivariant to the permutation symmetries of simple feedforward networks. However, they are not applicable to general architectures, since the permutation symmetries of a weight space can be complicated by recurrence or residual connections. This work proposes an algorithm that *automatically* constructs permutation equivariant models, which we refer to as *universal neural functionals* (UNFs), for any weight space. Among other applications, we demonstrate how UNFs can be substituted into existing learned optimizer designs, and find promising improvements over prior methods when optimizing small image classifiers and language models. Our results suggest that learned optimizers can benefit from considering the (symmetry) structure of the weight space they optimize. We open-source our library for constructing UNFs at https://github.com/AllanYangZhou/universal_neural_functional.

## 1 Introduction

Many problems in machine learning require handling *weight-space features*, such as the weights, gradients, or sparsity masks of neural networks. For example, optimizers iteratively map the current weights and gradient history to updated weights. Taking this perspective, researchers have proposed a variety of data-driven methods that train a neural network to process these weight-space features. Examples applications of these *neural functionals* [Zhou et al., 2023a] include training neural networks to predict classifier generalization from weights [Eilertsen et al., 2020], to optimize other networks [Metz et al., 2022], and to classify or edit implicit neural representations (INRs) [De Luigi et al., 2023].

Until recently, researchers lacked a unifying and principled framework for designing neural functionals, and would implement a custom model for their particular weight-space task. A significant recent advance was the development of weight-space models that are *permutation equivariant* [Navon et al., 2023, Zhou et al., 2023a]. *Neuron permutation* symmetries arise in a neural network's weight space because re-ordering hidden neurons has no effect on the network's function [Hecht-Nielsen, 1990]. A permutation equivariant neural functional can guarantee that under a neuron permutation of its input, its output permutes accordingly.

Navon et al. [2023] showed that permutation equivariance significantly improves performance on weight-space tasks, but their models only apply to the weight spaces of simple feedforward multilayer perceptrons (MLPs). Permutation equivariant neural functionals [Zhou et al., 2023a] added the ability to process weights from simple feedforward convolutional networks (CNNs). However, in practice we may deal with the weight spaces of complex networks that have residual connections, recurrence, normalization layers, and so on. Extending existing approaches to each possible weight space would be tedious and challenging.

38th Conference on Neural Information Processing Systems (NeurIPS 2024).

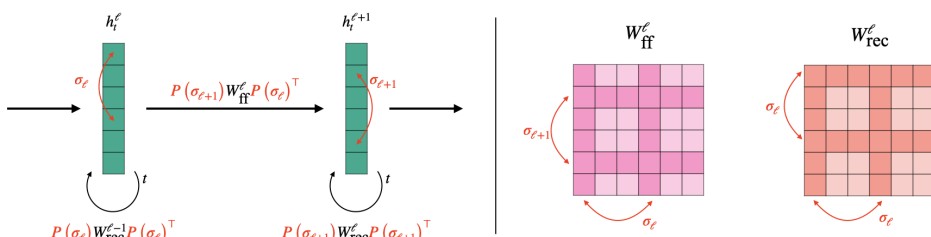

Figure 1: Illustration of the permutation symmetries in the weight space of a recurrent neural network (Example 2.2). **Left**: Each layer contains *feedforward* (ff) weights mapping between different layer's activations, and *recurrent* (rec) weights transforming activations over time. We can permute the hidden activations as illustrated without changing the final outputs $h_t^L$. **Right**: Permuting the hidden activations induces a permutation on the weights. Here, the rows and columns of the feedforward weights are permuted by $(\sigma_{\ell+1}, \sigma_\ell)$, while the recurrent weights are permuted by $(\sigma_\ell, \sigma_\ell)$. Our algorithm automatically constructs permutation equivariant models for any collection of weight tensors given a description of its symmetries (Appendix A).

We propose an approach that automatically constructs permutation equivariant models for *any* collection of tensors whose dimensions can permute according to a shared set of permutations. This naturally encompasses the permutation equivariance we might desire for any given weight space. We show that our algorithm constructs the most general linear layer that operates on a given weight space while guaranteeing equivariance to the specified permutation symmetries. Stacking multiple such layers with pointwise nonlinearities produces a deep permutation equivariant model, which we refer to as a *universal neural functional*.

To evaluate the empirical effectiveness of UNFs, we apply them to tasks that require processing networks with complex architectures containing recurrence, layer normalization, residual connections, and more. We use UNFs to implement learned optimizers and then optimize small image classifiers, RNNs, and Transformer language models, observing promising improvements over prior methods. In a generalization prediction task, we use UNF to predict the performance of sequence-to-sequence RNN models from their weights. Our experiments show that universal neural functionals are flexible, can be easily applied to different weight spaces, and improve upon prior weight-space methods.

## 2   Preliminaries

We largely follow or extend the notation and naming of Zhou et al. [2023a]. Given a fixed neural network architecture, there is a **weight space** $\mathcal{W}$ of possible parameters (weights, biases, normalization scalings, etc.). We refer to all such parameters as "weights". A particular set of weights $W = \left(W^{(1)}, \cdots, W^{(L)}\right)$ contains multiple "tensors", or multidimensional arrays. Depending on the architecture, $\mathcal{W}$ contains numerous symmetries [Hecht-Nielsen, 1990, Godfrey et al., 2022], i.e., transformations on the weight space that do not affect the network's behavior. Following prior work [Navon et al., 2023, Zhou et al., 2023a], this work focuses only on the permutation symmetries, which are called *neuron permutations*.

Neuron permutations correspond to re-arranging the neurons within (hidden) layers, which have no canonical ordering. We make the simplifying assumption that *all* layers can be re-arranged–this assumption can be later corrected using positional encodings [Zhou et al., 2023a]. Assuming there are $N$ *independently* permutable layers of neurons, the neuron permutation *group* is the direct product $\mathcal{S} = S_{n_1} \times \cdots \times S_{n_N}$, where $n_i$ is the number of neurons being permuted in each layer.

In general, each weight is a "tensor" (multi-dimensional array) of real numbers. Using $M(a, b, \cdots)$ to denote arrays $\mathbb{R}^{a \times b \times \cdots}$, consider a rank-$D_\ell$ tensor $W^{(\ell)} \in M\left(n_{d_1^\ell}, \cdots, n_{d_{D_\ell}^\ell}\right)$. Each dimension $d_i^\ell$ is permuted by $\sigma_{d_i^\ell}$. That is, the action of $\sigma$ on the indices of the weight tensor is:

$$\sigma\left(i_1, \cdots, i_{D_\ell}\right) := \left(\sigma_{d_1^\ell}(i_1), \cdots, \sigma_{d_{D_\ell}^\ell}(i_{D_\ell})\right). \tag{1}$$

Defining the multi-index $\vec{i} := (i_1, \cdots, i_{D_\ell})$, the action on the weight tensor is to permute the entries: $\left[\sigma W^{(\ell)}\right]_{\vec{i}} := W^{(\ell)}_{\sigma^{-1}(\vec{i})}$, and the action on $\mathcal{W}$ is $\sigma W := \left(\sigma W^{(1)}, \cdots, \sigma W^{(L)}\right)$.

We now elaborate on the definition of the group and action in several common cases.

**Example 2.1** (Multilayer perceptron). *A multilayer perceptron (MLP) with $L+1$ layers has activations $h^{\ell+1} = s\left(W^{(\ell)}h^\ell + b^{(\ell+1)}\right)$, with $h^1$ being the first (input) layer and $h^{L+1}$ the output. If each $h^\ell$ is a vector of length $n_\ell$, then the weights are matrices $W^{(\ell)} \in M(n_{\ell+1}, n_\ell)$ and the biases are vectors $b^{(\ell)} \in M(n_\ell)$. Then we have a neuron permutation group $\mathcal{S} = S_{n_1} \times \cdots \times S_{n_{L+1}}$, and $\sigma \in \mathcal{S}$ can be written $\sigma = (\sigma_\ell)_{\ell=1}^{L+1}$. The action on the weights and biases is:*

$$W^{(\ell)} \mapsto P(\sigma_{\ell+1}) W^{(\ell)} P(\sigma_\ell)^\top \quad and \quad b^{(\ell)} \mapsto P(\sigma_\ell) b^{(\ell)}, \tag{2}$$

*where $P(\sigma_\ell)$ is the $n_\ell \times n_\ell$ permutation matrix corresponding to $\sigma_\ell$. This corresponds exactly to the "NP" setting in Zhou et al. [2023a].*

**Example 2.2** (Recurrent neural network). *Consider a deep recurrent neural network (RNN) [Elman, 1990] without biases. We follow the presentation of Wang et al. [2023]:*

$$h_t^{\ell+1} = s\left(W_{rec}^{\ell+1} h_{t-1}^{\ell+1} + W_{ff}^\ell h_t^\ell\right), \tag{3}$$

*where $h_t^1$ are the inputs and $h_t^{L+1}$ are the outputs at each timestep, with $h_0^\ell$ initialized to $0$. The weight space consists of feedforward (ff) weights $W_{ff}^\ell \in M(n_{\ell+1}, n_\ell)$ and recurrent (rec) weights $W_{rec}^\ell \in M(n_\ell, n_\ell)$. We again define the neuron permutation group $S := S_{n_1} \times \cdots \times S_{n_{L+1}}$, but the action of the group on the weight space is now different. Here, re-arranging the neurons corresponds to transforming the weights:*

$$W_{ff}^\ell \mapsto P(\sigma_{\ell+1}) W_{ff}^\ell P(\sigma_\ell)^\top \quad and \quad W_{rec}^\ell \mapsto P(\sigma_\ell) W_{rec}^\ell P(\sigma_\ell)^\top.$$

*As illustrated by Figure 1, the feedforward weights transform just as in the MLP case (Eq. 2), but the recurrent weights' rows and columns must be transformed by the same permutation.*

**Example 2.3** (Convolutional neural network). *Consider a 1D convolutional neural network (CNN) without biases. Using $\star$ to denote cross-correlation, we have activations $h^{\ell+1} = s\left(W^{(\ell)} \star h^\ell\right)$, where the input is $h^1$ and the output is $h^{L+1}$. If each filter has spatial dimension $k_\ell$ and each $h^\ell$ has $n_\ell$ channels, then we have rank-3 weight tensors $W^{(\ell)} \in M(n_{\ell+1}, n_\ell, k_\ell)$ and neuron permutation group $\mathcal{S} = \prod_{\ell=1}^L S_{n_\ell} \times S_{k_\ell}$. Looking at how each dimension of $W^{(\ell)}$ permutes, we would have $\sigma_{n_{\ell+1}} \in S_{n_{\ell+1}}$ permute the first dimension (output channels), $\sigma_{n_\ell} \in S_{n_\ell}$ permute the second dimension (input channels), and $\sigma_{k_\ell} \in S_{k_\ell}$ permute the third dimension (spatial).*

*We note that permutating the spatial dimensions of a convolution filter would change the CNN's behavior and is not a true symmetry of the weight space. This is a notable difference between how our framework handles convolutional weight spaces compared to NFNs [Zhou et al., 2023a], where the action of the neuron permutation group does not affect the spatial dimensions at all. Assuming that all dimensions of each weight tensor can permute simplifies the development of our framework, and undesired symmetry can be broken (if desired) by positional encodings of the input [Zhou et al., 2023a, Lim et al., 2023].*

**Equivariance and invariance.** We are interested in functions $T : \mathcal{W} \to \mathcal{W}$ that are equivariant, meaning that it doesn't matter whether we apply a neuron permutation to the input or the output. We define $\mathbb{L}_\mathcal{S}(\mathcal{W}, \mathcal{W})$ as the space of equivariant linear maps, i.e., those $T$ satisfying:

$$T(\sigma W) = \sigma T(W), \forall \sigma \in \mathcal{S}, W \in \mathcal{W}. \tag{4}$$

Our goal is to design a layer (i.e., a parameterized space of functions) that is equivalent to $\mathbb{L}_\mathcal{S}(\mathcal{W}, \mathcal{W})$.

In some applications, we may instead desire invariance, that is a function $P$ satisfying

$$P(\sigma W) = P(W), \forall \sigma \in \mathcal{S}, W \in \mathcal{W}. \tag{5}$$

Following prior work [Navon et al., 2023, Zhou et al., 2023a], we can build invariant neural functionals by composing several equivariant layers with an invariant pooling layer, e.g., one that sums over every dimension of each weight tensor and concatenates the results.

# 3 Universal neural functionals

Since equivariance is preserved under composition, and pointwise non-linearities are already permutation equivariant, we can build deep equivariant models as long as we have an equivariant linear layer. Additionally, composing equivariant layers with an invariant pooling operation produces a deep invariant model. This section introduces a method for producing equivariant weight-space layers for any given weight space, which enables the flexible construction of *universal neural functionals*.

## 3.1 Decomposing equivariant weight-space maps

The weight space is a direct sum of individual weight subspaces $\mathcal{W} = \mathcal{W}^{(1)} \oplus \cdots \oplus \mathcal{W}^{(L)}$, so the problem of defining an equivariant layer on $\mathcal{W}$ can be decomposed into defining equivariant layers between each pair of weight subspaces $\mathcal{W}^{(m)}$ and $\mathcal{W}^{(\ell)}$, for all $\ell$ and $m$ [Navon et al., 2023].

We re-state this result in our own notation. For any $\ell, m$ pair we define $\mathbb{L}_{\mathcal{S}}\left(\mathcal{W}^{(m)}, \mathcal{W}^{(\ell)}\right)$ as the space of equivariant maps between the two weight subspaces. It contains all $T^{\ell m} : \mathcal{W}^{(m)} \to \mathcal{W}^{(\ell)}$ satisfying

$$T^{\ell m}\left(\sigma W^{(m)}\right) = \sigma T^{\ell m}\left(W^{(m)}\right) \quad \forall \sigma, W^{(m)}, \tag{6}$$

noting that the action on the left and right hand sides of the equivariance condition are not, in general, the same.

Assume that we already have a basis $\mathcal{B}^{sp}$ for $\mathbb{L}_{\mathcal{S}}\left(\mathcal{W}^{(p)}, \mathcal{W}^{(s)}\right)$. A basis function $E \in \mathcal{B}^{sp}$ can be extended to $\bar{E} : \mathcal{W} \to \mathcal{W}$ by defining:

$$\bar{E}(W)^\ell := \begin{cases} E\left(W^{(p)}\right) & \ell = s \\ 0 & \text{otherwise} \end{cases}, \tag{7}$$

where $\bar{E}(W) := \left(\bar{E}^1(W), \cdots, \bar{E}^L(W)\right)$.

**Theorem 3.1** (Navon et al. [2023])**.** *Let $\left\{\mathcal{B}^{\ell m}\right\}$ be bases for each $\mathbb{L}_{\mathcal{S}}\left(\mathcal{W}^{(m)}, \mathcal{W}^{(\ell)}\right)$. Then the union of these bases (extended by Eq. 7) is a basis for linear equivariant maps on $\mathcal{W}$. That is, we have the basis $\mathcal{B}$ for $\mathbb{L}_{\mathcal{S}}\left(\mathcal{W}, \mathcal{W}\right)$ defined:*

$$\mathcal{B} = \bigcup_{\ell, m \in [\![L]\!]^2} \left\{ \bar{E} \mid E \in \mathcal{B}^{\ell m} \right\}. \tag{8}$$

This result tells us that we can construct an equivariant basis $\mathcal{B}$ for $\mathbb{L}_{\mathcal{S}}\left(\mathcal{W}, \mathcal{W}\right)$ by simply combining the equivariant bases $\left\{\mathcal{B}^{\ell m}\right\}$ for each pair of weight subspaces.

## 3.2 Equivariant layers between tensors

Since weights are tensors, our decomposed problem involves finding bases for permutation equivariant maps between tensors. Variants of this problem have been studied by numerous prior works–in particular, Maron et al. [2018] theoretically characterize a basis for equivariant maps between arbitrary-rank tensors, and provide a concrete implementation of the basis functions in the rank-2 case. Here, we describe a *general* algorithm that automatically constructs a basis for permutation equivariant maps between arbitrary-rank tensors. Concretely, it implements each basis function in terms of simple array operations that are amenable to efficient computation with modern deep learning frameworks.

---
**Algorithm 1** Basis for equivariant $\mathcal{W}^{(m)} \to \mathcal{W}^{(\ell)}$ layer

---
**Require:** $\mathcal{W}^{(m)}, \mathcal{W}^{(\ell)}$
1: Initialize basis $\mathcal{B}^{\ell m} \leftarrow \{\ \}$
2: $\mathcal{I} \leftarrow \left\{ o_1, \cdots, o_{D_\ell}, i_1, \cdots, i_{D_m} \right\}$
3: **for** $\mathcal{P}$ in VALIDPARTITIONS $(\mathcal{I})$ **do**
4:     Label each subset $s_p \in \mathcal{P}$ by unique character CHAR$(s_p)$
5:     **for** $\alpha \in \mathcal{I}$ **do**
6:         Map index $c[\alpha] \leftarrow$ CHAR$(s_p)$ where $\alpha \in s_p$
7:     **end for**
8:     $E_{\mathcal{P}}(X)_{c[o_1], \cdots, c[o_{D_\ell}]} := \sum_{\mathcal{R}} X_{c[i_1], \cdots, c[i_{D_m}]}$
9:     $\mathcal{B}^{\ell m} \leftarrow \mathcal{B}^{\ell m} \cup \left\{ E_{\mathcal{P}} \right\}$
10: **end for**
11: return $\mathcal{B}^{\ell m}$

---

Functions in $\mathbb{L}_\mathcal{S}\left(\mathcal{W}^{(m)}, \mathcal{W}^{(\ell)}\right)$ take input tensors indexed by $\{\, i_1, \cdots, i_{D_m}\,\}$ and produces output tensors indexed by $\{\, o_1, \cdots, o_{D_\ell}\,\}$. We can construct a basis $\mathcal{B}^{\ell m}$ for this space where each element is identified by a **valid partition** $\mathcal{P}$ of these indices. Recall that the indices $(i_1, i_2, \cdots)$ of $W^{(m)}$ are permuted by $\left(\sigma_{d_1^m}, \sigma_{d_2^m}, \cdots\right)$. We say that two indices $i_1$ and $i_2$ "permute simultaneously" if $d_1^m = d_2^m$.

**Definition 1.** *A **valid partition** is a partition $\mathcal{P}$ of the output and input indices $\mathcal{I} = \left\{\, o_1, \cdots, o_{D_\ell}, i_1, \cdots, i_{D_m}\,\right\}$ into non-empty subsets, such that each subset only contains indices that are permuted simultaneously.*

**Example 3.1** ($\mathcal{W}^{(m)} = \mathcal{W}^{(\ell)} = \mathbb{R}^{n_1 \times n_2}$). *Here the output and input indices are $\{\, o_1, o_2, i_1, i_2\,\}$. The partition $\{\, \{\, o_1, o_2\,\}, \{\, i_1, i_2\,\}\,\}$ is **not** valid because $o_1, o_2$ are permuted by $\sigma_1, \sigma_2$, so they do not permute simultaneously. On the other hand, $\{\, \{\, o_1, i_1\,\}, \{\, o_2, i_2\,\}\,\}$ is a valid partition.*

**Example 3.2** ($\mathcal{W}^{(m)} = \mathcal{W}^{(\ell)} = \mathbb{R}^{n_1 \times n_1}$). *This time, the partition $\{\, \{\, o_1, o_2\,\}, \{\, i_1, i_2\,\}\,\}$ is valid because $o_1, o_2$ are both permuted by $\sigma_1$, as are $i_1, i_2$.*

To construct the equivariant basis, we enumerate all valid partitions and then map each partition $\mathcal{P}$ to a basis function $E_\mathcal{P}$. Concretely, we label each subset of $\mathcal{P}$ with a distinct character $\alpha, \beta, \gamma, \cdots$ and then remap each of our original indices $\{\, o_1, \cdots, o_{D_\ell}, i_1, \cdots, i_{D_m}\,\}$ to a a character based on which subset the index was in. This mapping is best illustrated by continuing our previous example.

**Example 3.3** ($\mathcal{W}^{(m)} = \mathcal{W}^{(\ell)} = \mathbb{R}^{n_1 \times n_2}$). *Here input and output are both matrices, with combined indices $\{\, o_1, o_2, i_1, i_2\,\}$. We have two permutations $(\sigma_1, \sigma_2) \in S_{n_1} \times S_{n_2}$ that can act on the rows and columns of the input and output matrices. There are four valid partitions:*

$$\mathcal{P}_1 = \{\, \{\, o_1, i_1\,\}, \{\, o_2, i_2\,\}\,\}, \qquad \mathcal{P}_2 = \{\, \{\, o_1, i_1\,\}, \{\, o_2\,\}, \{\, i_2\,\}\,\},$$
$$\mathcal{P}_3 = \{\, \{\, o_1\,\}, \{\, i_1\,\}, \{\, o_2, i_2\,\}\,\}, \qquad \mathcal{P}_4 = \{\, \{\, o_1\,\}, \{\, o_2\,\}, \{\, i_1\,\}, \{\, i_2\,\}\,\}. \qquad (9)$$

*Consider $\mathcal{P}_2$–we assign a character to each subset:*

$$\mathcal{P}_2 = \{\, \underbrace{\{\, o_1, i_1\,\}}_{\alpha}, \underbrace{\{\, o_2\,\}}_{\beta}, \underbrace{\{\, i_2\,\}}_{\gamma}\,\}. \qquad (10)$$

*which tells us to remap the output indices $(o_1, o_2) \mapsto (\alpha, \beta)$ and the input indices $(i_1, i_2) \mapsto (\alpha, \gamma)$, producing the basis function $E_{\mathcal{P}_2}\left(W^{(m)}\right)_{\alpha\beta} := \sum_\gamma W^{(m)}_{\alpha\gamma}$, where summation over $\gamma$ can be inferred because it only contains an input index.*

*Repeating this index-remapping process for each valid partition will generate a total of four basis functions $E_{\mathcal{P}_1}, \cdots, E_{\mathcal{P}_4}$ for $\mathbb{L}_\mathcal{S}\left(\mathcal{W}^{(m)}, \mathcal{W}^{(\ell)}\right)$. Our equivariant $\mathcal{W}^{(m)} \to \mathcal{W}^{(\ell)}$ layer will be defined as the linear combination $T^{\ell m}\left(W^{(m)}; \lambda\right) := \sum_{k=1}^4 \lambda_k \cdot E_{\mathcal{P}_k}\left(W^{(m)}\right)$, which is the layer introduced in Hartford et al. [2018].*

To generalize the previous example, for each valid partition of the indices $\mathcal{P}$ we label its subsets with characters $\alpha, \beta, \gamma, \cdots$ and then construct a basis function:

$$E(W^{(m)})_{c[o_1], \cdots, c[o_{D_\ell}]} = \sum_\mathcal{R} W^{(m)}_{c[i_1], \cdots, c[i_{D_m}]}, \qquad (11)$$

where $c[\cdot]$ maps each index to the subset of $\mathcal{P}$ that contains it. We sum over the characters in $\mathcal{R}$, which is the (possibly empty) subset of characters that only contain input indices (i.e., only appear on the right-hand side). Entries that are not explicitly assigned by the left-hand side are 0. Algorithm 1 gives a formal description of the complete process for generating $\mathcal{B}^{\ell m}$.

**Theorem 3.2.** *Algorithm 1 produces a basis for the equivariant linear maps from $\mathcal{W}^{(m)}$ to $\mathcal{W}^{(\ell)}$.*
**Proof.** *See Appendix B.1.*

Once Algorithm 1 has generated a basis of equivariant functions $\mathcal{B}^{\ell m}$, we can implement an equivariant layer using a vector $\lambda^{\ell m} \in \mathbb{R}^{|\mathcal{B}^{\ell m}|}$ of learned coefficients:

$$T^{\ell m}\left(W^{(m)}; \lambda^{\ell m}\right) := \sum_{b=1}^{|\mathcal{B}^{\ell m}|} \lambda_b^{\ell m} \cdot E_{\mathcal{P}_b}\left(W^{(m)}\right). \qquad (12)$$

### 3.3 Equivariant layers on weight spaces

Theorem 3.1 now tells us that we may now construct the equivariant weight-space layer by combining the bases $\{\mathcal{B}^{\ell m}\}$ into a basis $\mathcal{B}$ of functions on $\mathcal{W}$. The weight-space layer $T(\cdot, \lambda)$ can then be defined by a linear combination of the basis functions with learned coefficients $\lambda$. Explicitly, the full layer is defined:

$$T(W, \lambda) = \left( T^1\left(W, \lambda^{1,:}\right), \cdots, T^L\left(W, \lambda^{L,:}\right) \right), \tag{13}$$

where $\lambda^{\ell,:} = \{ \lambda^{\ell m} \mid \ell = 1, \cdots, L \}$ and $T^\ell\left(W, \lambda^{\ell,:}\right) = \sum_{m=1}^{L} T^{\ell m}\left(W^{(m)}, \lambda^{\ell m}\right)$.

Appendix A provides a concrete description of how we specify the weight space in code and how the algorithm is then used to automatically construct an equivariant weight space layer. Our open-source implementation is compatible with most JAX [Bradbury et al., 2018] neural network libraries.

**Theorem 3.3.** *The weight-space layer (Eq.-13) is $\mathcal{S}$-equivariant, and can express any linear equivariant function on $\mathcal{W}$.*

**Proof.** *Each $T^{\ell m}$ is a linear combination of basis functions in $\mathcal{B}^{\ell m}$. Then, as described by Thm 3.1, Eq. 13 is a linear combination of functions that form a basis for $\mathbb{L}_\mathcal{S}(\mathcal{W}, \mathcal{W})$.*

For an MLP weight space with neuron permutation group defined as in Example 2.1, this approach will generate the exact same layer as NFN$_{\text{NP}}$ [Zhou et al., 2023a]. This is because the layers each parameterize all possible linear maps equivariant to the same symmetry group, and hence can express the same set of functions.

### 3.4 Multiple feature channels

In practice, we may be interested in simultaneously processing multiple weight-space features, such as the weights and a history of gradients. These features can be stacked into a "channel" dimension analogous to the channels of convolutional networks. In that case, we must consider direct sums of weight spaces of the form $\mathcal{W}^c = \oplus_{k=1}^{c} \mathcal{W}$, with elements that can be written as[1] $W = (W[1], \cdots, W[c])$, for $W[k] \in \mathcal{W}$. Then the action is $\sigma W := (\sigma W[1], \cdots, \sigma W[c])$ for $\sigma \in \mathcal{S}$, extending the (single channel) definition. The definition of equivariance can then be extended to layers of the form $T(\cdot) : \mathcal{W}^{c_i} \to \mathcal{W}^{c_o}$, where $c_i, c_o$ are the number of input and output channels.

Extending equivariant layers to the multi-channel setting is quite common in the geometric deep learning literature and simply involves taking linear combinations along the channel dimension [Cohen and Welling, 2016, Ravanbakhsh et al., 2017]. That is, we modify the equivariant layer between subspaces as:

$$T^{\ell m}\left(W^{(m)}; \lambda^{\ell m}\right)[k'] := \sum_{b=1}^{|\mathcal{B}^{\ell m}|} \sum_{k=1}^{c_i} \lambda_b^{\ell m}[k', k] \cdot E_{\mathcal{P}_b}\left(W^{(m)}\right)[k], \tag{14}$$

where each $\lambda_b^{\ell m}$ is now a learned $c_o \times c_i$ matrix instead of a scalar.

### 3.5 Deep models

The previous sections describes the construction of $\mathcal{S}$-equivariant layers that operate operate on weight-space features in $\mathcal{W}^c$. We construct *universal neural functionals* by stacking multiple such layers (interleaved with pointwise non-linearities) into a deep, permutation equivariant model that can process weights. To construct a permutation invariant model, we can add an invariant pooling layer after the equivariant layers, as in prior work [Navon et al., 2023, Zhou et al., 2023a].

## 4 Experiments

In this section, we refer to weight-space models constructed using our algorithm as **universal neural functionals (UNFs)**. We compare their performance to prior methods on two types of weight-space tasks: predicting the generalization of recurrent sequence-to-sequence models, and training learned optimizers for a variety of architectures and datasets.

---

[1]In the multichannel setting we overload notation and use $W$ to refer to elements of $\mathcal{W}^c$, not $\mathcal{W}$.

## 4.1 RNN generalization prediction

One promising application of neural functionals is in predicting the generalization of neural network models from their weights Eilertsen et al. [2020]. We construct **Tiny RNN Zoo**[2], a dataset of recurrent neural networks trained to do arithmetic by completing given questions character-by-character. For example, given the input string "15+20=" the correct completion would be "35<EOS>". To construct the dataset, we train $10^4$ sequence-to-sequence [Sutskever et al., 2014] models on example problems with input numbers up to five input digits. Both encoder and decoder RNNs contain a single GRU cell [Chung et al., 2014] with hidden size 128. Each model is trained with a distinct learning rate and batch size, and it's test success rate (SR) is recorded. The learning rate is sampled from a log-uniform distribution over $[10^{-4}, 10^{-2}]$, and the batch size is sampled uniformly from $\{64, 128, 256\}$. With the goal of predicting test SR from weights, we split the Tiny RNN Zoo into 8000/1000/1000 training, validation, and test examples.

The success rate of each RNN model is clearly invariant under permutation symmetries of its weights, so invariance is a natural inductive bias for any generalization predictor. We evaluate STATNN [Unterthiner et al., 2020] and a UNF-based predictor (note that NFNs are not applicable to the weights of recurrent networks). STATNN is operates on basic statistical features[3] of the weights, and has been shown to be a very strong baseline on previous generalization prediction tasks [Unterthiner et al., 2020]. On the other hand, UNF operates on raw weight inputs and may be able to extract more nuanced signals than STATNN, as was shown (for CNN classifiers) in Zhou et al. [2023a].

| Method | Test $\tau$ |
|---|---|
| Deep Set | $0.8306 \pm 0.0006$ |
| STATNN | $0.8839 \pm 0.0007$ |
| **UNF (Ours)** | **$0.8968 \pm 0.0006$** |

Table 1: Rank correlation between predicted and actual success rates of RNNs on an arithmetic task. Predicting with UNF significantly outperforms STATNN [Unterthiner et al., 2020].

In particular, STATNN computes the mean, variance, and $(0, 25, 50, 75, 100)$-percentiles of each weight tensor in the RNN and feeds them into a six-layer MLP with hidden width 600. UNF is a permutation invariant model, implemented using a three-layer equivariant backbone (16 hidden channels) followed by invariant pooling and a three-layer MLP (512 hidden neurons). We train each predictor with binary cross entropy loss (since the target SR is in $[0, 1]$), using the Adam optimizer with learning rate 0.001, batch size 10, and training for up to 10 epochs. We use the validation data only for early stopping, and assess the performance of each predictor on the test inputs using Kendall's $\tau$, the rank correlation between predicted and actual success rate.

**Results.** Table 1 shows the performance of each predictor on held out weight inputs. Our UNF-based predictor achieves significantly higher rank correlation between predicted and actual success rate, suggesting that the equivariant layers are able to extract more informative features from the raw weights compared to STATNN.

## 4.2 Learned optimizers

Choosing the optimizer is a key step in training any modern neural network. Though most popular optimizers are variants of stochastic descent, the non-convexity of neural network training leaves few rigorous guidelines for ideal optimizer design. This has led some researchers to propose *training* good optimizers using some form of meta-learning [Bengio et al., 1990, 2013, Andrychowicz et al., 2016, Wichrowska et al., 2017, Metz et al., 2019].

Common optimizers today (including the learned ones) are equivariant to any permutation of the weights. This is because permuting the weights also permutes the gradients, so stochastic gradient descent and similar optimizers will produce permuted updates. However, equivariance to *any* permutation ignores the actual symmetry structure of the optimized neural network. Arguably the more appropriate constraint is to only require equivariance to the *neuron permutation group*, which enables more expressive optimizers while still respecting the symmetries of the weight space. As we will see, this can be achieved by using UNFs to implement a learned optimizer.

Training learned optimizers that generalize well is extremely compute-intensive [Metz et al., 2022], so we conduct our experiments in several smaller settings to analyze the impact of architecture

---

[2]Inspired by the Tiny CNN Zoo [Unterthiner et al., 2020].
[3]Notably, it computes statistics that are invariant to permutations of the weights.

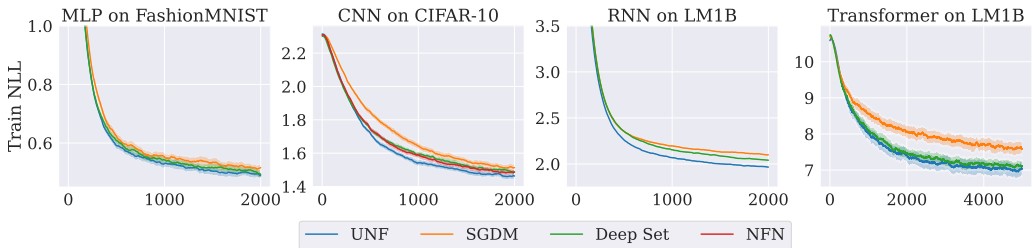

Figure 2: Training loss (negative log-likelihood) curves for different tasks and architectures using meta-learned optimizers. We implement learned optimizers with either universal neural functionals (**UNF**s), **NFN**s [Zhou et al., 2023a], or **Deep Sets** [Zaheer et al., 2017]. Deep Sets are the current standard choice for implementing learned optimizers. Note that NFN is identical to UNF in the MLP case, different for CNN case, and not applicable to RNNs or Transformers. All loss curves are smoothed and averaged over 5 random initializations (3 for Transformer), with shaded regions showing standard error.

choice on learned optimizer performance. In each setting, an optimizer is meta-trained to optimize an architecture type on a task from random initializations. Following Harrison et al. [2022], our learned optimizers track momentum terms $m_t^\gamma \leftarrow \gamma m_{t-1} + \nabla_t$ and produce updates of the form:

$$W_{t+1} \leftarrow W_t - \alpha \left( m_t^{\gamma_0} + \beta f \left( W_t, \nabla_t, \{ m_t^{\gamma_i} \}_i, t \right) \right). \tag{15}$$

Here $\alpha m_t^{\gamma_0}$ is a "nominal term" that biases the learned optimizer to behave like stochastic gradient descent with momentum coefficient $\gamma_0$. The neural functional $f(\cdot)$ ingests weights $W_t$, gradients $\nabla_t$, momentum terms at several coefficients $\{ m_t^{\gamma_i} \}_i$, and the iteration $t$.

During meta-training, we optimize network $f$ and scalars $\alpha, \beta, \gamma_0$ to minimize the task training loss after a fixed number of training steps $T$, the "inner training horizon." To avoid the issue of backpropagating through an optimization process, we estimate meta-gradients using persistent evolutionary strategies [Vicol et al., 2021].

**Comparisons.** The default architecture choice for $f(\cdot)$ in prior work is **Deep Sets** [Zaheer et al., 2017], which offers equivariance to *any* permutation symmetry. We study the effect of replacing Deep Sets by **UNF**s. We also try the **NFN_{NP}** architecture [Zhou et al., 2023a] where applicable, though it cannot be used on the RNN and Transformer experiments. Finally, we consider stochastic gradient descent with momentum (**SGDM**), which is equivalent to fixing $\beta = 0$ in Eq. 15. The SGDM baseline is also meta-trained to tune the learning rate $\alpha$ and momentum decay rate $\gamma_0$. We compare the different learned optimizers in four tasks:

**MLP on FashionMNIST.** Each optimizer trains an MLP classifier on a downsized and flattened version of the FashionMNIST dataset [Xiao et al., 2017]. We note that for MLP weight spaces, UNF are identical to NFN_{NP} [Zhou et al., 2023a].

**CNN on CIFAR-10.** Each optimizer trains a convolutional classifier on a downsized $16 \times 16$ CIFAR-10. In this setting our algorithm produces a UNF that is *different* to NFN_{NP} (see Example 2.3).

**RNN on LM1B.** Each optimizer trains a character-level RNN-based language model (LM) on the One Billion Word Language Model Benchmark (LM1B) dataset [Chelba et al., 2013].

**Transformer on LM1B.** Each optimizer trains a Transformer LM on LM1B, this time predicting tokens instead of characters.

We use an inner training horizon $T = 2{,}000$ for the first three tasks and $T = 5{,}000$ for the Transformer task, since it takes longer to train. When implementing $f(\cdot)$ for each method, we use a network with four layers, 32 hidden channels, and ReLU nonlinearities. The Deep Set optimizer uses exclusively Deep Set layers [Zaheer et al., 2017, Eq. 4], while the UNF and NFN optimizers uses three Deep Set layers followed by a single UNF or NFN layer. See Appendix C.1-C.2 for full descriptions of the tasks and meta-training.

**Results.** Figure 2 shows the training curves produced by each of the meta-trained optimizers in each experiment. Learned optimizers with deep architectures (UNF, Deep Set, or NFN) outperform SGDM, even after tuning SGDM's learning rate and momentum decay. UNF typically learns fastest

and achieves the lowest training loss across all methods, though Deep Set and NFN can be comparable in some settings. One interesting observation is that UNF outperforms NFN in the CNN experiment. As noted in Example 2.3, UNFs make the stronger assumption that all tensor dimensions–including the spatial dimensions of the convolution filter–are permutable, while NFNs do not. Although the UNF assumption is technically incorrect, the stronger assumption leads to a lower parameter count (see Table 3 in the appendix) which may be easier for meta-optimization.

Overall, our results show the promise of using UNFs to create more expressive learned optimizers that utilize the specific symmetry structure of the weight spaces they optimize. Further work could investigate their capacity for generalization to new tasks and architectures, for example by meta-training on diverse tasks [Metz et al., 2022]. Moreover, as Table 3 in the appendix shows, a necessary trade-off of UNFs being more expressive is that they require more parameters for an equivalent number of layers and hidden channels. Since learned optimizers are still much smaller than the networks they could optimize, this may not be a significant computational constraint in practice. Still, it could be a challenge to meta-optimization, since evolutionary strategies are known to struggle in higher dimensions. Hence, further work on efficient high-dimensional meta-gradient estimators would complement the development of expressive weight-space models like UNF.

## 5 Related Work

There is a long history of neural network architectures that are equivariant to various symmetry groups [LeCun et al., 1995, Cohen and Welling, 2016, Ravanbakhsh et al., 2017, Kondor and Trivedi, 2018, Cohen et al., 2018]. Existing frameworks for automatically constructing equivariant models [Finzi et al., 2021] produce equivariant matrices, which would be intractable for our task. Our work constructs efficient equivariant basis functions for a particular class of permutation symmetries that arise in the weight spaces of neural networks. Permutation equivariant networks have been developed for sets [Zaheer et al., 2017], matrices whose rows and columns permute independently [Hartford et al., 2018], and tensors under *higher-order* permutation actions [Thiede et al., 2020, Pan and Kondor, 2022]–the latter may also be viewed as equivariant models on graphs or polytopes [Maron et al., 2018, Albooyeh et al., 2019]. This work observes that a weight space is a *collection* of tensors under higher-order permutation symmetries, and develops equivariant models for that setting.

There has been significant interest in designing architectures that that either optimize or generate neural network weights [Schmidhuber, 1993, Ha et al., 2016, Krueger et al., 2017, Kirsch and Schmidhuber, 2021, Peebles et al., 2022, Metz et al., 2022]. Some works have identified the importance of respecting the relevant symmetries when implementing black box meta-learners [Kirsch et al., 2022]. However, precise characterizations of equivariant models on neural weight spaces are relatively recent and were initially restricted to simple feedforward models [Navon et al., 2023, Zhou et al., 2023a,b].

A recent alternative approach has been to leverage message passing neural networks (MPNNs) [Zhang et al., 2023] to process weights as edges of a graph. Concurrent to this work, Kofinas et al. [2024] demonstrated applications of MPNNs to learned optimization for MLPs and CNNs and Lim et al. [2023] extended MPNNs to process general weight-spaces. MPNN-based approaches benefit from more flexible adaptation to heterogenous inputs, and the computational cost of message passing does not grow as rapidly as our basis–this is because our approach guarantees each linear layer to be maximally expressive while MPNNs do not. We give a more detailed exposition of this trade-off in Appendix B.3

## 6 Conclusion

We introduce a method for constructing permutation-equivariant neural functionals that operate on arbitrary weight spaces, removing a major limitation of previous frameworks that were only applicable to the weight spaces of simple MLPs and CNNs. Our algorithm constructs maximally expressive equivariant linear layers for processing any collection of tensors given a description of their permutation symmetries, and implements these layers in terms of efficient array operations in standard deep learning frameworks. We empirically validate that the resulting *universal neural functionals* (UNFs) are effective at tasks that involve processing the weights and gradients of convolutional image classifiers, recurrent sequence-to-sequence models, and Transformer language models. In

particular, we find that UNFs show promising improvements over existing learned optimizer designs in small scale experiments.

**Limitations and future work.** It remains to be demonstrated how UNFs can be applied to heterogenous weight-space inputs, e.g., to have a single UNF act as a learned optimizer for any input architecture. Moreover, our experimental results only validate the promise of UNF-based learned optimizers in relatively limited settings, and more work would needed to test generalization across arbitrary tasks. Finally, computational tractability may be a significant challenge for more complex architectures as the number of basis terms generated by Alg. 1 would grow rapidly for higher rank tensors with higher-order interactions (see Appendix B.2). Resolving these challenges would further improve the scalability and applicability of neural functionals to weight-space tasks.

## 7   Acknowledgements

We thank Jascha Sohl-Dickstein and Yiding Jiang for insightful general discussions about the project, and Louis Kirsch for helpful feedback on early drafts. AZ is supported by the NSF Graduate Research Fellowship Program. We are grateful to the TPU Research Cloud (TRC) for providing compute for some of the experiments.

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

## A Weight-space specifications

Here we discuss the concrete **specification** that precisely describes a weight space and must be provided as input to the algorithm before it can construct equivariant weight-space layers. Our implementation is compatible with most JAX [Bradbury et al., 2018] neural network libraries.

Suppose we wish to process an MLP's weights that are stored in a (nested) Python dictionary:

```
params = {
    "layer1": {"weight": Array[64, 32], "bias": Array[64]},
    "layer2": {"weight": Array[64, 64], "bias": Array[64]},
}
```

Then a specification should match the nested dictionary structure but provide a string or integer name for each dimension of each array. The name tells the algorithm which permutation affects which dimensions of each array.

In this example, the specification closely follows the MLP description in Example 2.1, where $W^{(1)} \in M(n_2, n_1)$ is permuted as $W^{(1)} \mapsto P(\sigma_2) W^{(1)} P(\sigma_1)^\top$.

```
specification = {
    "layer1": {"weight": ("n2", "n1"), "bias": ("n2",)},
    "layer2": {"weight": ("n3", "n2"), "bias": ("n3",)},
}
```

Providing this `specification` object to our algorithm is sufficient for it to deduce the symmetry group, its action, and construct the corresponding equivariant layer.

Since most neural networks consist of repeating layers or blocks, the process of constructing the specification can be semi-automated by first defining a function that creates the specification for a single layer or block and then re-using that function for each block. Although we did not find this necessary for our experiments, it may also be possible to automatically deduce the specifications for a network in common deep learning frameworks by analyzing its computation graph.

## B Further analysis of UNFs

### B.1 Algorithm 1 generates a basis for $\mathbb{L}_\mathcal{S}\left(\mathcal{W}^{(m)}, \mathcal{W}^{(\ell)}\right)$

Here we show that Algorithm 1 produces a basis $\mathcal{B}^{\ell m}$ for $\mathbb{L}_\mathcal{S}\left(\mathcal{W}^{(m)}, \mathcal{W}^{(\ell)}\right)$, the space of linear equivariant maps between $\mathcal{W}^{(m)}$ and $\mathcal{W}^{(\ell)}$. Consider instantiating these linear maps as matrices multiplying flattened input $\text{vec}\left(W^{(m)}\right)$. Maron et al. [2018] characterize a basis $\left\{ B^\mu \right\}_\mu$ for these matrices, where the entries of each basis matrix are defined:

$$B_{a,b}^\mu = \begin{cases} 1 & (a,b) \in \mu \\ 0 & \text{otherwise} \end{cases}. \tag{16}$$

Here $a \in \mathbb{I}_m$ and $b \in \mathbb{I}_\ell$ are multi-indexes for the input and output spaces, and $\mu \in \mathbb{I}_m \times \mathbb{I}_\ell / \sim$ is an equivalence class of the combined input-output index space $\mathbb{I}_m \times \mathbb{I}_\ell$ under the equivalence relation $\sim$ defined by $a \sim a'$ if and only if $a_i = a_j \iff a_i' = a_j'$ for all $i, j$, i.e. the two multi-indexes $a, a'$ have the same *equality pattern*.

Re-arranging Maron et al. [2018, Eq. 10b], any equivariant linear map is defined:

$$L(W^{(m)})_b = \sum_{a \in \mathbb{I}_m} \sum_{\mu \in \mathbb{I}_m \times \mathbb{I}_\ell / \sim} w_\mu B_{a,b}^\mu W_a^{(m)} = \sum_{\mu \in \mathbb{I}_m \times \mathbb{I}_\ell / \sim} w_\mu \sum_{a \in \mathbb{I}_m} I\{(a,b) \in \mu\} W_a^{(m)}, \tag{17}$$

where $I\{\cdot\}$ is an indicator function for the given condition.

Notice that each equivalence class $\mu$ is represented by what we call a *valid partition* of $[D_m + D_\ell] := \{1, \cdots, D_m + D_\ell\}$, so this is already a sum over valid partitions as in Eq. 12. We can now observe

that each term on the RHS is equivalent to one of our basis functions (Alg 1 Line 8). That is, for a given equivalence class $\mu$ represented by valid partition $\mathcal{P}$:

$$\sum_a I\{(a, b) \in \mu\} W_a^{(m)} = E_{\mathcal{P}}(W^{(m)}). \tag{18}$$

This is because for any $\mathcal{I} := (a, b)$ yielding a nonzero term on the LHS, if $i, j \in [D_m + D_\ell]$ are grouped together by partition $\mathcal{P}$ then $\mathcal{I}_i = \mathcal{I}_j$, otherwise they would violate the *equality pattern* of $\mu$. Therefore, we can replace all indices grouped together in a partition with a single shared symbol, i.e. the characters in Eq. 11.

Hence, Algorithm 1 produces a basis that spans the same space of equivariant functions defined in Maron et al. [2018], but constructs the basis functions in terms of efficient array operations instead of as matrices. Note that this is similar to the construction in Pan and Kondor [2022], but generalized to multi-node sets (non-square tensors whose axes can potentially permute independently).

## B.2 Size of basis produced by Algorithm 1

Suppose we have a neuron permutation symmetry group $S = S_{n_1} \times \cdots \times S_{n_N}$, i.e., every neuron permutation $\sigma$ is composed of $N$ distinct permutations $(\sigma_1, \cdots, \sigma_N)$. For each $i = 1, \cdots, N$ we define $c_i\left(\mathcal{W}^{(\ell)}\right)$ to be the number of indices that $\sigma_i \in S_{n_i}$ permutes in weight tensors $W^{(\ell)} \in \mathcal{W}^{(\ell)}$ (which could be 0). Finally, denote $b(k)$ to be the k'th Bell number. Then the number of basis functions generated by Algorithm 1 is:

$$|\mathcal{B}^{\ell m}| = \sum_{\ell, m} \prod_{i=1}^{N} b\left(c_i\left(\mathcal{W}^{(\ell)}\right) + c_i\left(\mathcal{W}^{(m)}\right)\right). \tag{19}$$

## B.3 Comparison to MPNN-based approaches

Each UNF layer can express any linear equivariant function on a given weight space (Thm 3.3). Compared to methods based on message-passing neural networks (MPNNs), this means UNFs can have very expressive individual layers, but may also be more computationally challenging due to the growth in the size of the basis (see next section).

As an example, consider a simple "RNN" where $h_{t+1} = W h_t$ and $h_t \in \mathbb{R}^n$ has exchangeable entries, meaning that $W \mapsto PWP^T$ is a symmetry. Algorithm 1 would produce an equivariant basis with $b(2+2) = 15$ terms[4].

On the other hand, we could construct a parameter graph [Lim et al., 2023] with $n$ nodes and $2n^2$ directed edges between them (allowing a forward and backward edge for each weight, equivalently $n^2$ undirected edges). Then using a similar construction to Lim et al. [2023, Appendix C.1.2], we would get a linear GNN that computes:

$$f(W) = aW_{\star, \star} + bW_{j, \star} + cW_{\star, k} + dW_{k, \star} + eW_{\star, j} + fW_{jk}, \tag{20}$$

which is a linear combination of 6 equivariant basis functions, instead of 15. This leads to a potientially interesting trade-off between expressivity vs tractability. However, we also note that in practice MPNNs use non-linear MLPs in their message passing updates, and the comparison between UNF and MPNN-style approaches remains an open empirical question.

# C  Experimental details

## C.1  Learned optimization tasks

Here we describe each of the experimental settings we evaluated the learned optimizers on. Across all experiments, the training loss is negative log-likelihood.

---

[4]In this case, the full basis is also given by Maron et al. [2018, Appendix A].

Figure 3: Number of parameters used by $f(\cdot)$ in each learned optimizer, for each task. Note that NFN and UNF are identical for the MLP task. This count does not include the other meta-learned scalars in Eq. 15, which are $\alpha, \gamma_0, \beta$.

| Task | UNF | Deep Set | NFN |
|------|-----|----------|-----|
| MLP on FashionMNIST | 3,783 | 2,788 | 3,783 |
| CNN on CIFAR-10 | 7,369 | 2,788 | 41,603 |
| RNN on LM1B | 8,043 | 2,788 | N/A |
| Transformer on LM1B | 64,168 | 2,788 | N/A |

**MLP on FashionMNIST.** Train a three-layer MLP classifier on a downsized ($8 \times 8$) and flattened version of the FashionMNIST dataset [Xiao et al., 2017]. The MLP has a hidden size of 32 and ReLU activation function. We use a batch size of 128.

**CNN on CIFAR-10.** Train a convolutional classifier on a downsized $16 \times 16$ CIFAR-10. The classifier has two convolutional layers (16 and 32 channels), followed by global average pooling and a linear classification head, and is trained with a batch size of 128.

**RNN on LM1B.** Trains a character-level RNN-based language model (LM) on LM1B [Chelba et al., 2013]. The RNN itself has one hidden layer with size 64, and uses identity-initialization [Le et al., 2015]. An embedding layer with dimension 32 maps tokens to embeddings before feeding into the RNN, and an output layer produces token predictions from the RNN output. The LM is trained to predict the next token with teacher forcing at batch size 64, on sequences of length 16.

**Transformer on LM1B.** Train a Transformer LM on LM1B, this time predicting tokens instead of characters. The Transformer has two blocks with an embedding dimension of 32, and uses four self-attention heads. We train with a batch size of 8 on length-8 sequences.

## C.2 Learned optimization meta-training

Call `DS[c]` a single equivariant Deep Set layer [Zaheer et al., 2017, Eq 4] with $c$ output channels (similarly for `UNF[c]` and `NFN[c]`). Then $f(\cdot)$ in our learned optimizers (Eq. 15) is always implemented as a feedforward architecture:

```
DeepSetOpt = DS[32] -> ReLU -> DS[32] -> ReLU -> DS[32] -> ReLU -> DS[1]
UNFOpt = DS[32] -> ReLU -> DS[32] -> ReLU -> DS[32] -> ReLU -> UNF[1]
NFNOpt = DS[32] -> ReLU -> DS[32] -> ReLU -> DS[32] -> ReLU -> NFN[1]
```

For all methods, we initialize $\alpha = 0.1$ and $\gamma_0 = 0.9$ before starting meta-training. For non-SGDM methods, we initialize $\beta = 0.001$, and provide six momentum values $\{ m_t^{\gamma_i} \}_i$ with coefficients $\gamma_i = 0.1, 0.5, 0.9, 0.99, 0.999, 0.9999$. The iteration number $t$ is converted into an 11-dimensional sinusoidal encoding, and all inputs to $f(\cdot)$ are concatenated along the channel dimension. Concretely, this results in an input in $\mathcal{W}^{19}$. The output is in $\mathcal{W}^1$.

We meta-train for 50,000 steps using Adam, estimating meta-gradients over 16 parallel training runs using persistent evolutionary strategies (PES) [Vicol et al., 2021] with a truncation length of 50 and a noise standard deviation of 0.01. The meta-training objective is training loss at the end of the inner training horizon ($T = 5,000$ for the Transformer setting, and $T = 2,000$ otherwise), and we apply a gradient clipping of 1.0.

**Size of each learned optimizer** $f(\cdot)$**.** Since Deep Set layers are agnostic to the specific weight space being optimized, the Deep Set learned optimizer uses the same number of parameters in each task. The same is not true of UNF layers, where the number of parameters grows in proportion to the size of the bases generated by Algorithm 1. Table 3 lists the number of parameters in $f(\cdot)$ for each learned optimizer.

## C.3 Compute

Experiments were run on a mix of TPU v3 and v4 accelerators. On a TPU v3-8, training a UNF for our RNN generalization prediction task takes $< 3$ hours. Also on a TPU v3-8, meta-training a UNF

for one of our learned optimizers takes $\sim 4$ hours for the MLP task, $\sim 7$ hours for the CNN task, and $\sim 20$ hours for the RNN task.

