# OpenReview forum: "Universal Neural Functionals"
_NeurIPS.cc/2024/Conference — NeurIPS 2024 poster_

### Official Review · Reviewer_fP4N · 2024-06-21

**Soundness:** 4
**Presentation:** 3
**Contribution:** 3
**Rating:** 7
**Confidence:** 3

**Summary:**

This paper develops an algorithm for constructing functions that take neural network parameters / parameter-derived values as inputs but are equivariant / invariant to the inherent permutation symmetries in neural network weights. More specifically, this work is applicable to a larger class of neural network architectures than previous approaches (including RNNs), due to the less restrictive assumptions made. The authors evaluate their method on two types of task: 1) as a model for predicting generalisation from neural network weights, 2) as a learnable optimiser for training neural networks more effectively than traditional methods like SGD. They find moderate improvements with their method over previous approaches.

**Strengths:**

- The approach is well-motivated from a mathematical perspective on permutation equivariant / invariant functions, and the experiments appear to remain faithful to the proposed methodology rather than resorting to an assortment of ad-hoc hacks / tricks.
- The paper is on the whole very well written. In particular, the authors do a good job of explaining the mathematics that, although not conceptually too difficult, is inherently cumbersome / fiddly.
- The work generalises previous work to a wider class of neural networks, rather than being specific to just a couple of architectures, and is therefore a good contribution to the area (I am not very familiar with the related work so I am relying on the authors' own summary of the related work to make this judgement).

**Weaknesses:**

- The RNN generalisation prediction experiment only compares against one baseline, which is not far behind the proposed method. It is therefore difficult to judge the significance of these numbers.
- The authors state "Navon et al. [2023] showed that permutation equivariance significantly improves performance on weight-space tasks, but their models only apply to the weight spaces of simple feedforward multilayer perceptrons (MLPs)." but do not appear to test any methods without permutation symmetries when performing their experiments. In particular, it would have been nice to see whether a learned optimiser that doesn't account for permutations symmetries (e.g. a plain-old neural network?) performs significantly worse than the proposed method or DeepSet, which also is designed to take permutation symmetries into account.
- There are some small changes I would make to the paper for readability, please see my suggestions below.

**Questions:**

## Questions
- In Example 3.1 we seem to be assuming that weights from 2 separate layers ($m$ and $\ell$) will be permuted with the same permutation because they have the same size. Surely we can permute two layers on neurons in different ways? Please clarify as I may be missing something here.
- On line 182, could you expand on what is meant by "Entries that are not explicitly assigned by the left-hand side are 0."?
- In Eq. 15 are we manually computing the momentum term still, and only using the learned optimiser to produce the new direction / update at each step?
- You mention that your assumption of permutation symmetry over the spatial dimension in CNNs is technically incorrect, but that this can be addressed somehow by use of positional encodings. Did you use this positional encoding fix in your experiments?
- My understanding of the UNF is that it is technically not a neural network, as might be common in other work on learned optimisers, but rather we learn coefficients for a particular set of basis functions that you have derived, however the computations we end up needing are quite similar to neural networks and can therefore be implemented using standard deep learning libraries. Is this correct?

## Suggestions
- In the second paragraph of the Preliminaries section, circa line 64, it would be useful to explicitly say "where $S_n$ is the symmetric group of $n$ elements", "$\sigma$ is a particular permutation in this group", "the usual / canonical action of $\sigma$ on the set of indices permutes them according to the permutation $\sigma$". Clarifying explanations like these do not take up much space but vastly improve the readability of the paper.
- On line 72, it does not appear that $s$ is defined anywhere, but appears to be the non-linearity of the neural network. A short note would improve clarity.
- Typo on line 246: Remove the word "is" from "STATNN is operates on basic statistical features of the weights"

**Limitations:**

- The authors discuss some general limitations of the current algorithm, such as computational cost and applicability to more general problems, though these are generally beyond the scope of the current paper.

---

> ### Author Rebuttal · Authors · 2024-08-07
>
> We thank the reviewer for their thoughtful comments.
>
> > The RNN generalisation prediction experiment only compares against one baseline, which is not far behind the proposed method. It is therefore difficult to judge the significance of these numbers.
>
> We believe this scale of improvement is typical in the literature for generalization prediction; despite its simplicity StatNN is a very strong baseline. For example, in “GNNs for learning equivariant representations” (Kofinas et al (ICLR 2024 oral)), the best method NG-T outperformed StatNN by only ~0.02 on rank correlation. As suggested by a reviewer, we also ran a Deep Set comparison, which significantly underperforms both our method and StatNN.
>
> Test rank correlation between predicted and actual performance (higher is better):
>
> | Method     | Kendall's tau       |
> |------------|---------------------|
> | Deep Set   | $0.8306 \pm 0.0006$ |
> | STATNN     | $0.8839 \pm 0.0007$ |
> | UNF (Ours) | $0.8968 \pm 0.0006$ |
>
> > do not appear to test any methods without permutation symmetries when performing their experiments. In particular, it would have been nice to see whether a learned optimiser that doesn't account for permutations symmetries (e.g. a plain-old neural network?) performs significantly worse than the proposed method or DeepSet, which also is designed to take permutation symmetries into account.
>
> This would be an interesting experiment to run, but to our knowledge a non-equivariant method (such as a simple neural network ingesting all weights) would be extremely computationally costly to run in learned optimization due to the high dimensionality of weights. For this reason, we are not aware of learned optimizers in the literature that use this type of (non-equivariant) architecture. Instead, prior works prefer to use per-parameter MLPs, as you mentioned.
>
> However, for completeness, we are currently running an RNN generalization prediction experiment with a non-equivariant baseline (a simple MLP ingesting all the weights) and will update when the results are available.
>
> > In Example 3.1 we seem to be assuming that weights from 2 separate layers ($m$ and $\ell$) will be permuted with the same permutation because they have the same size
>
> Good point, two weights having the same size does not necessarily mean their dimensions are permuted the same way. This is just a limitation of our notation--$\mathbb{R}^{n_1 \times n_2}$ is our way of saying that permutation $\sigma_1$ affects dimension 1 and permutation $\sigma_2$ affects dimension 2. For a matrix with the same size but with different permutations, we might write it as belonging to $\mathbb{R}^{n_3 \times n_4}$, even if $n_3=n_1$ and $n_4 = n_2$. Unfortunately, we are not aware of a better notation for conveying this point, but we will clarify it in the text.
>
> > On line 182, could you expand on what is meant by "Entries that are not explicitly assigned by the left-hand side are 0."?
>
> Equation 11 is assigning values to entries of the tensor $E(W^{(m)})$, but in general the indices generated by the characters on the LHS would not cover all the possible indices of the tensor. So any indices *not* specified by Equation 11 are assumed to be 0.
>
> > In Eq. 15 are we manually computing the momentum term
>
> The momentum terms fed as input are computed manually / in the standard way.
>
> > Did you use this positional encoding fix in your experiments?
>
> We did not find positional encoding necessary in this case, no. We will clarify the text on this
>
> > My understanding of the UNF is that it is technically not a neural network [...] but rather we learn coefficients for a particular set of basis functions that you have derived [...]
>
> Your statement "we learn coefficients for a particular set of basis functions that you have derived" is correct, and moreover the learned linear combination of basis functions forms a single "layer" that we stack with nonlinearities that we stack and optimize much like a neural network, which is why we refer to it as one. One can say that its layers are different from that of common neural networks, but also the layers of a convolutional network are different from that of an RNN.
>
> > There are some small changes I would make to the paper for readability, please see my suggestions below.
>
> Thank you for catching these issues and providing suggestions, we will include them in the updated manuscript.

---

> > ### Comment · Reviewer_fP4N · 2024-08-08
> > **Response to rebuttal**
> >
> > Thank you for your response.
> >
> > > As suggested by a reviewer, we also ran a Deep Set comparison
> >
> > > for completeness, we are currently running an RNN generalization prediction experiment with a non-equivariant baseline (a simple MLP ingesting all the weights) and will update when the results are available.
> >
> > These extra baselines will be very helpful for contextualising the results of the paper. This was my primary criticism of the paper, so it's nice to see it addressed, even if all it took was adding a couple more baselines.
> >
> > I also thank the authors for clarifying my points of confusion in the paper. Hopefully small changes / additions to the paper can be made so that other people will not run into these same comprehension issues.
> >
> > Other reviewers have raised some interesting points, but my evaluation of the paper remains positive, and so I shall be maintaining my score of 7 (accept).

---

### Official Review · Reviewer_7uT6 · 2024-07-05

**Soundness:** 3
**Presentation:** 3
**Contribution:** 2
**Rating:** 5
**Confidence:** 3

**Summary:**

The paper proposes Universal Neural Functionals that are models operating on neural network weights in a permutation equivariant way. Compared to previous works, UNFs are more general and can be more easily applied to architectures such as RNNs and Transformers. The UNFs show good results in generalization prediction for RNNs and also improve learned optimizers in several image and language tasks.

**Strengths:**

1. The paper addresses the problem of learning from network weights which is challenging and has a lot of potential.
2. Interesting learning to optimize experiments, which is a very relevant application of this kind of method.
3. A method is inspired by the symmetry group theory, which is an interesting alternative to message passing and graph networks and potentially could be very powerful in this task.
4. The paper is well written and organized.

**Weaknesses:**

1. The algorithm 1 that constructs equivariant weight-space layers need "specifications" of a network that indicate which dimensions of which layers are permuted simultaneously. While for simple architectures like MLPs providing them is easy, for architectures like RNN and Transformer it seems tricky and essentially requires digging into the exact implementation of each layer (e.g. how the weights in the multi-head attention layers are split in heads and q, k, v weights). It would be useful to see the specifications for RNNs and Transformers similarly to the ones for MLP in Appendix A. Given that the results for Transformers in Fig. 2 are not very strong, it may indicate that the specifications are incorrect. So the paper is a bit misleading in a sense that it claims to "automatically construct permutation equivariant models for any weight space", but it does not automatically constructs the specifications which is actually a quite tricky part for some complicated models with complicated implementations and "analyzing its computation graph" may not be enough since it usually reveals the connections between layers, but not the connections between dimensions.

2. The theory of the paper is a bit disconnected from the experiments making it harder to understand the theory part and its strengths. How Algorithm 1 works in practice for the networks in the experiments? How the valid partition looks like for those networks? How the basis looks like and are there any insights about the architecture given the basis (e.g. how the basis will look like for mlps, cnns, rnns and transformers)?

3. RNN generalization prediction

- More baselines are needed. For example, Deep Sets could be one simple baseline, also since the architecture of the RNN models is the same, a simple MLP on concatenated parameters can be another baseline.
- it's questionable that 0.8968 is significantly better than 0.8839.
- Given those two weaknesses, I believe the setup should be changed to include more diverse architectures to highlight UNF's strength. For example, see the Graph Metanetworks paper for the inspiration on how to construct "Diverse Architectures" experiments.

4. Learned optimizers

- L289: it's questionable that Deep Sets is "the default architecture choice for f". For example, in the paper "Practical tradeoffs between memory, compute, and performance in learned optimizers" (Metz et al, 2022) a simple per-parameter MLP (small_fc_lopt) is used that does not include deep set layers. So directly using that or similar optimizer as one of the baselines would be interesting.
- Why Adam is not used instead of SGDM as the backbone of learned optimizers in Eq. 15 given that Adam is a standard choice for optimizing Transformers used in the experiments?
- In CNN on CIFAR-10, why UNF is better than NFN-NP given that UNF assumes spatial dimensions can be permuted? Is positional encoding added to spatial dimensions in UNF?

5. Related work is missing some relevant papers like "Hyper-Representations as Generative Models: Sampling Unseen Neural Network Weights" and "NeRN - Learning Neural Representations for Neural Networks".

Overall, given that there are very few experiments, lack of empirical analysis and the results are often similar to the baselines, I'm inclined towards rejection.

**Questions:**

For SGDM do the authors actually learn α and γ0 or it's simply tuned?

**Limitations:**

the limitations are discussed

---

> ### Author Rebuttal · Authors · 2024-08-07
>
> We thank the reviewer for their detailed feedback. We will also include the suggested relevant works in our discussion.
>
> > it does not automatically constructs the specifications which is actually a quite tricky part for some complicated models
>
> We agree that constructing specifications can be tricky, and we will include the specifications for our RNNs and Transformers. Analyzing the computation graph to automatically deduce the specifications is an interesting idea and may be possible if we first specify how each of the primitive operations use the dimensions of their parameter tensors. For example, a Linear layer implemented as $Y=XW$ always connects the first dimension of its parameter W to the second dimension of the previous layer's parameter.
>
> > How Algorithm 1 works in practice for the networks in the experiments? How the valid partition looks like for those networks?
>
> Thanks for raising this very interesting point. Although the bases in general can look quite complex, we can already analyze specific differences in the bases generated for processing MLPs and RNNs. For example, valid partitions of the kind we give in Example 3.2 ($\mathcal{W}^{(m)}=\mathcal{W}^{(\ell)}=\mathbb{R}^{n_1 \times n_1}$) appear for RNNs but not for MLPs, because in RNNs the outputs at one timestep are used as the inputs for the next timestep. Hence we have the same permutation action on the input space and output space of recurrent weights. We will expand our analysis of these differences in the paper to better connect theory and practice.
>
> > RNN generalization prediction: It's questionable that 0.8968 is significantly better than 0.8839.
>
> This scale of improvement is typical in the literature for generalization prediction; despite its simplicity StatNN is a very strong baseline (see the additional comparison we just ran again Deep Sets). For example, in “GNNs for learning equivariant representations” (Kofinas et al (ICLR 2024 oral)), the best method NG-T outperformed StatNN by only ~0.02 on rank correlation.
>
> > RNN generalization prediction:  More baselines are needed.
>
> As suggested, we have added a Deep Sets (Zaheer et al, 2017) comparison. As the results below show, our method (UNF) and StatNN (Unterthiner et al, 2020) outperform the method based on Deep Sets by a wide margin.
>
> Test rank correlation between predicted and actual performance (higher is better):
>
> | Method     | Kendall's tau       |
> |------------|---------------------|
> | Deep Set   | $0.8306 \pm 0.0006$ |
> | STATNN     | $0.8839 \pm 0.0007$ |
> | UNF (Ours) | $0.8968 \pm 0.0006$ |
>
> > it's questionable that Deep Sets is "the default architecture choice for f".
>
> To clarify, what we call the "Deep Set" baseline in our learned optimization experiments is actually the per-parameter MLP used in [Metz et al, 2022](https://arxiv.org/abs/2203.11860). This is because if you exclude the $\gamma$ term in Deep Sets (Zaheer et al, Eq 4) and stack multiple layers, the result is equivalent to a per-parameter MLP. We apologize for the confusion and will clarify the text. Regardless, the actual experiments we ran are in fact comparing with the standard architecture choice for learned optimizers, such as the one used by Metz et al, 2022.
>
> > For SGDM do the authors actually learn α and γ0 or it's simply tuned?
>
> We learn these hyperparameters using the same ES optimizer as we use for all the other learned optimizer results.
>
> > Why Adam is not used instead of SGDM as the backbone of learned optimizers in Eq. 15 given that Adam is a standard choice for optimizing Transformers used in the experiments?
>
> The main purpose of the learned optimizer experiments is to demonstrate the impact due to architecture for learned optimizers with the same backbone, in this case SGDM. However, we are currently running experiments with Adam as the backbone as well.
>
> > In CNN on CIFAR-10, why UNF is better than NFN-NP given that UNF assumes spatial dimensions can be permuted? Is positional encoding added to spatial dimensions in UNF?
>
> This is a good question--we did not find it necessary to add positional encoding for the spatial positions in UNF. One explanation is that spatial parameter sharing makes UNF more parameter-efficient, which can be helpful for ES optimizers like the ones used for learned optimization, since they can struggle at very large (high dimensional) parameter spaces. We will expand on this point in our experimental discussion.

---

> > ### Comment · Reviewer_7uT6 · 2024-08-12
> >
> > I appreciate reviewers' response, which address my concerns, therefore I raise the score.

---

### Official Review · Reviewer_k71j · 2024-07-11

**Soundness:** 3
**Presentation:** 3
**Contribution:** 2
**Rating:** 5
**Confidence:** 4

**Summary:**

The paper proposes Universal Neural Functionals (UNFs), which are models that process the weights of other neural networks. UNFs are equivariant to permutation symmetries of neurons and applicable to general architectures. The authors formalize neural network weights as sets of tensors, and develop an algorithm that constructs maximally expressive equivariant linear layers for processing any collection of tensors given a description of their permutation symmetries. The resulting UNFs outperforms StatNN, a baseline that uses statistical features, on an RNN generalization prediction task. Additionally, UNFs improves over several other architectures on learning optimizers for various image classifiers and language models.

**Strengths:**

- UNF has the appealing property of being applicable to general architectures. Plus, the code appears easy to use on custom neural networks. This work is thus likely to be useful for the deep learning community.
- The algorithm that computes maximally expressive equivariant linear layers for a set of tensors is a significant theoretical contribution.
- Connection to past work is discussed well throughout the paper.
- The paper is generally clearly written and well organized. The examples in Section 2 and 3 are particularly helpful in showing the wide range of architectures UNFs are applicable on and in helping readers understand the concept of valid partitions.
- Both experiments uses datasets with decent size compared with related works and demonstrates promising performance of UNFs.

**Weaknesses:**

- Despite the claims in the abstract and conclusion, there has been other work, most notably Lim et al. 2023, that developed permutational equivariant weight-space models applicable to general architectures. This unfortunately weakens the novelty of this paper. The narrative might need to be modified to emphasize other contributions, which are still significant, such as constructing maximally expressive equivariant linear layers and promising results for learned optimizers.
- I find Section 3 a bit difficult to follow. Part of the reason could be tensor indices are inherently complicated. However, it would be helpful if the authors could provide intuition along with stating results. For example, why do valid partitions produce a basis and other partitions do not?
- It is not clear whether the proposed architecture scales well, since the number of valid partitions can be very large especially when many indices permute simultaneously or there are many layers.
- In both experiments, the proposed method is not compared to the most recent permutation equivariant architectures (Zhang et al. 2023, Kofinas et al. 2024, Lim et al. 2023). These papers are highly relevant, as they solve the same category of problems (processing or generating neural network weights) and feature similar advantages (being permutational equivariant, and Lim et al. 2023 also has the ability to process arbitrary weight spaces). Additionally, they have been published on line more than two months before NeurIPS’s deadline so are not generally considered contemporaneous.

**Questions:**

- Among the many applications of weight-space networks, how did the authors decide on the two tasks to conduct experiment on?
- In line 316, it is stated that UNF makes the stronger assumption than NFN that all tensor dimensions are permutable, but line 273-276 seem to suggest the UNF respect fewer symmetries. Could the authors clarify?

**Minor issues / suggestions**
- $\sigma_{d^l_i}$ and $\sigma$ in line 67 are not defined.
- In Equation (1) and the two lines after it, $\sigma$ depends on $l$, unless one assumes all weights have the same dimensions. This dependency should be made more explicit.
- In Example 2.1, is the set of weight matrices $\{W_1, …, W_{L}\}$? If so, shouldn’t the indices of $S$ and $\sigma$ range from 1 to $L$ instead of $L+1$?
- Since Theorem 3.1 uses basis for linear equivariant maps, $\mathbb{L}$ defined in line 119 should be the space of linear equivariant maps, instead of the space of equivariant maps.
- Line 165 typo “a a character”
- What is $W$ on the right side of the equation in line 172?
- Line 538 “next section” -> previous section

**Limitations:**

The authors include an informative limitation section that points out challenges and future directions.

---

> ### Author Rebuttal · Authors · 2024-08-07
>
> We thank the reviewer for their careful analysis and feedback. We will incorporate the suggestions and fixes proposed here.
>
> > Despite the claims in the abstract and conclusion, there has been other work …
>
> We agree that the novelty of this work lies in the construction of maximally expressive equivariant layers, and in empirical results on learned optimizers for complex architectures like RNNs and Transformers. We will update the abstract, intro, and conclusion to emphasize this aspect of our contribution and also address the contribution of related graph-network approaches.
>
> >  it would be helpful if the authors could provide intuition along with stating results. For example, why do valid partitions produce a basis and other partitions do not?
> We agree that the explanation can be difficult to follow due in part to dealing with indices for arbitrary-rank tensors. One justification for the definition of valid partitions is given in the proof in Appendix B.1: the valid partitions can be identified with equivalence classes of indices (a,b) that are used to define the basis matrices (Eq (16)).
>
> For intuition, we may also work out parameter sharing patterns for simple permutation equivariant layers. Following [Ravanbakhsh et al](https://arxiv.org/abs/1702.08389), we can find the parameter sharing for a layer by studying the orbits of the input and output index spaces. By studying a few examples (such as the one we give in Example 3.3) one observes that each orbit must be identified with a valid partition of the indices. We will expand our examples to include this intuition.
>
> >  the number of valid partitions can be very large especially when many indices permute simultaneously or there are many layers
>
> We agree that this can be a limitation, and characterize the exact growth of the number of basis functions in Eq 19. Our goal and theoretical contribution here was to characterize maximally expressive equivariant layers, i.e., including all possible basis terms. It is an interesting area of future research to consider whether one could select a good subset of the basis functions that perform well in practice, while also being computationally cheaper.
>
> > the proposed method is not compared to the most recent permutation equivariant architectures
>
> We will attempt to include comparisons to these architectures for the updated manuscript, though full comparisons are challenging because, to our knowledge, Lim et al does not provide source code and Kofinas et al have not yet published their learned optimization code for direct comparison.
>
> > how did the authors decide on the two tasks to conduct experiment on?
>
> Learned optimization is a natural and challenging task for methods that can learn to edit the weights of other networks, and would also potentially have plenty of downstream impact. Generalization prediction is also an interesting weight-space task studied in prior relevant work (Navon et al, 2023, Zhou et al 2023).
>
> Although many past weight-space papers also included experiments on INRs, we omit INR experiments in this paper because all the INRs involved in those experiments were actually MLPs, whereas the focus of our work is to extend to processing more general architectures.
>
> >  it is stated that UNF makes the stronger assumption than NFN that all tensor dimensions are permutable, but line 273-276 seem to suggest the UNF respect fewer symmetries
>
> Good point, L273-276 was only meant when comparing UNF to Deep Sets (the standard architecture for learned optimizers until recently), not NFNs. We will clarify the text on this point.
>
> > What is $W$ on the right side of the equation in line 172?
>
> Should be the input $W^{(m)}$, thank you for catching this
>
> > In Example 2.1, is the set of weight matrices $W_1, \cdots, W_L$? If so, shouldn’t the indices of $S$ and $\sigma$ range from $1$ to $L$ instead of $L+1$
>
> Not quite, a feedforward network with $L$ weight matrices has $L+1$ layers of neurons, when we include the input and output layers. And neurons are the things that give rise to our permutation symmetries. For example, consider 2 weight matrices $W_1,W_2$. There are three layers of neurons: input, hidden, and output.

---

> > ### Comment · Reviewer_k71j · 2024-08-11
> >
> > Thank you for the response, which clarifies most of my questions. My only remaining reservation is the lack of comparison, especially empirical ones, to several recent works that propose permutational equivariant weight-space models for general architectures (Zhang et al. 2023, Kofinas et al. 2024, Lim et al. 2023). I am maintaining my rating for now.

---

### Official Review · Reviewer_J2cE · 2024-07-13

**Soundness:** 3
**Presentation:** 2
**Contribution:** 2
**Rating:** 5
**Confidence:** 2

**Summary:**

Extending from recent works, the authors propose a new neural network layer architecture that enforces permutation equivariance in the weight space. The proposed architecture is used for learned optimizers and compared with other existing methods in a series of tasks, archiving improvement over the state-of-the-art methods.

**Strengths:**

1. Methodology is well-developed.
2. Tests on various architectures show improvement over state-of-the art methods.

**Weaknesses:**

1. The presentation of the methodology is not very clear (specifically around Eq. 9 and 10).
2. The presentation of the results is also not very clear and the results are not sufficiently reported.
3. Whilst the training times for the test cases are reported, the inference speed is not reported.

**Questions:**

See weaknesses. Please improve the presentation of methodology and results.

**Limitations:**

The authors have adequately addressed the limitations.

---

> ### Author Rebuttal · Authors · 2024-08-07
>
> We thank the reviewer for their review and suggestions, which we have incorporated into the draft manuscript.
>
> > presentation of the methodology is not very clear (specifically around Eq. 9 and 10)
>
> We apologize for any confusion, and have polished the presentation throughout, including more intuitive explanations for Eq. 9 and 10 and also expanded examples. If there are any specific points of confusion to address, we would also welcome more detailed comments.
>
> > the results are not sufficiently reported
>
> We have run additional experiments in response to various requests from other reviewers--we welcome any additional specific feedback on the presentation or reporting of the results.

---

> > ### Comment · Reviewer_J2cE · 2024-08-11
> >
> > Thanks for the reply. I will keep my score.

---

### Official Review · Reviewer_eHaW · 2024-07-17

**Soundness:** 3
**Presentation:** 4
**Contribution:** 4
**Rating:** 7
**Confidence:** 2

**Summary:**

The study develops permutation equivariant models, developing an algorithm for defining a permutation eqivariant map of tensors with arbitrary rank, applying them in training learned optimizations and generalization prediction and finds that these class of models have improved performance on weight space tasks. This algorithm can be adapted to residual networks and transformers, a substantive selling point of this work.

**Strengths:**

The paper is pretty strong, showing good results for tasks which are traditionally hard to work on.

The method is novel, and applicable for various weight space tasks.

The methodology is well introduced and explained well.

**Weaknesses:**

line 219, operate is mentioned twice.

There is a limited number of datasets evaluated upon.

I would like to see a table showing the computational efficiency/time of this method vs other methods.

**Questions:**

Can you produce a table of computational speed + resource comparisons?

---

> ### Author Rebuttal · Authors · 2024-08-07
>
> We thank the reviewer for their thoughtful assessment--we are glad that the review recognized the novelty of the work and the challenging nature of the problem space.
>
> > There is a limited number of datasets evaluated upon.
>
> We will expand the evaluation in a few ways, based on suggestions from reviewers. First, we added a Deep Sets baseline for the RNN generalization experiment, and found that UNF continues to perform best (see table of test rank correlations below). Additionally, we are currently generating generalization prediction datasets for other architectures such as Transformers, and will include those results once available.
>
> Rank correlation between predicted and actual performance (higher is better):
>
> | Method     | Kendall's tau       |
> |------------|---------------------|
> | Deep Set   | $0.8306 \pm 0.0006$ |
> | STATNN     | $0.8839 \pm 0.0007$ |
> | UNF (Ours) | $0.8968 \pm 0.0006$ |
>
> > Can you produce a table of computational speed + resource comparisons?
>
> Appendix Section C.3 currently contains information about the computational costs of running our UNF methods, we will update it to also include computational cost numbers for the baselines.

---

### Decision · Program_Chairs · 2024-09-25

**Decision:**

Accept (poster)

**Comment:**

The paper presents a novel approach to permutation equivariant models, developing an algorithm for creating permutation equivariant maps of tensors, which is applied to weight space tasks with promising results. The strengths of the study include its innovative methodology, which is well-explained and has been successfully applied to various neural network architectures, including RNNs and Transformers. The paper is well-organized and clearly written, with a strong theoretical foundation that is supported by experiments on a range of architectures, demonstrating improvements over state-of-the-art methods. Reviewers' concerns are properly addressed.